# Association of Nursing Work Environment, Relationship with the Head Nurse, and Resilience with Post-Traumatic Growth in Emergency Department Nurses

**DOI:** 10.3390/ijerph18062857

**Published:** 2021-03-11

**Authors:** Sun-Young Jung, Jin-Hwa Park

**Affiliations:** Research Institute of Nursing Science, College of Nursing, Daegu Catholic University, Daegu 42472, Korea; syjung@cu.ac.kr

**Keywords:** emergency nursing, leadership, posttraumatic growth, Republic of Korea, resilience, work environment

## Abstract

Emergency department nurses are confronted with unpredictable diseases and disasters and work-related traumatic stress events. This study aimed to examine the relationship between nursing work environment, relationship with the head nurse, resilience, and posttraumatic growth among emergency department nurses. Data were collected from December 2018 to February 2019 through a self-administered survey questionnaire. Participants comprised 127 nurses working in the emergency department. The collected data were analyzed using *t*-test, analysis of variance with Scheffé’s test, Pearson’s correlations, and hierarchical multiple regression. The mean posttraumatic growth score of emergency department nurses was 2.59 ± 0.64 out of a possible 5.00. The posttraumatic growth showed a statistically significant difference according to age in the emergency department nurse. Resilience was the most significant variable controlling other variables, accounting for 29% of the variability. The findings support that intervention programs should be developed to encourage a positive relationship with the head nurse and enhance resilience in emergency department nurses.

## 1. Introduction

Today, our society is seeing an increasing number of new infectious diseases and unpredictable disasters, which can lead to a spike in the number of emergency patients. The emergency room is a place where immediate primary care is provided to these patients by experienced healthcare professionals [1]. Regardless of their work, nurses in emergency departments are more likely to be exposed to a variety of trauma cases, in addition to verbal and physical violence from some patients and their guardians, doctors, or colleagues [2]. In addition, they encounter direct or indirect experiences of traumatic events such as pain, death, or the dying process of patients at the front line while responding to ensure customer satisfaction emphasized by the hospital management. Such trauma experiences among emergency department nurses have been shown to lead to posttraumatic stress [3] with empathetic fatigue [4]. In fact, a previous study shows that approximately one-third of emergency department nurses are at high risk for posttraumatic stress disorder [5]. The World Health Organization defined coronavirus disease 2019 as a pandemic in 2020, and according to the report, more than 35,000 medical staff members were infected or died [6]. In such situations, emergency department nurses are particularly vulnerable to exposure to infectious agents and traumatic stress [7].

Trauma-related studies of emergency department nurses centered on posttraumatic stress disorders [2,8] found that nurses did not experience only negative aspects when experiencing various forms of trauma [9]. Rather, nurses develop posttraumatic growth, wherein they experience positive emotions while saving patients’ lives and improving their outcomes [9]. Such posttraumatic growth develops a new meaning of life or deeper spiritual beliefs through cognitive efforts during trauma; therefore, it has been said that posttraumatic growth and posttraumatic stress coexist [10].

The nursing work environment is a comprehensive concept of dynamic relationships, which includes the relationship with professional and qualified members and organizational characteristics of hospitals to provide high-quality professional nursing services [11]. In addition, nurses’ critical thinking and clinical judgment are shaped and developed by the nursing environment [12]. In previous research, the nursing work environment was identified as an influencing factor that reduced negative nursing outcomes such as exhaustion, turnover, and job dissatisfaction [13]. However, only a few studies have been conducted so far to confirm that the nursing work environment is a factor associated with posttraumatic growth for emergency room nurses.

In addition, the nursing work environment in the emergency department is a busy and complex environment in which various professional staff members have both horizontal and vertical relationships [14]. Leadership, response/teamwork, and resourcing are especially important to provide high-quality care in this nursing environment [15], and in this organizational system, the head nurse is the most frequent contact in the vertical relationships of emergency department nurses. Although there has been no study on the relationship between posttraumatic growth and relationship with the head nurse, a previous study focusing on the relationship between leaders and members showed the perceived quality of exchange relations of staff nurses with head nurses is a more important factor influencing nurses’ job satisfaction and organizational commitment than the quality of exchange relations of head nurses [16].

There are individual differences in response to stress, and there is an increasing recognition that even when experiencing stress, it is possible to overcome it with individual strengths and abilities. Emergency departments are confronted with rapidly changing situational factors or emergency situations with a lot of posttraumatic stress. In such situations, resilience can occur in such adversity [17] and be presumed to be the most necessary individual quality for nurses. Resilience is a multisystemic construct and the positive development and functionality under atypical stress [18]. It helps nurses alleviate the effects of stress and difficulties at work [19]. In addition, resilience has an impact on improving nursing work performance by facilitating effective coping with crisis situations during work [20]. Although no study has examined the association between resilience and posttraumatic growth in emergency department nurses, it was found that there was no significant correlation between resilience and posttraumatic growth in Israeli mental health nurses [21]. However, among vocational school nursing students, moderate psychological resilience appears to be associated with the highest level of posttraumatic growth [22]. Therefore, further research on the relationship between resilience and posttraumatic growth is necessary. Resilience, a process between different promotive and protective factors of multiple systems [18], is thought to be an important factor that can enhance adaptation even in stressful situations such as traumatic events [23]. Accordingly, this study intended to provide basic data for understanding posttraumatic growth and identify the association with the nursing work environment of emergency room nurses, their relationship with the head nurse, resilience, and posttraumatic growth.

Therefore, the purpose of this study was to understand the association of emergency room nurses’ work environment, relationship with the head nurse, and resilience with posttraumatic growth. With this purpose, the hierarchical regression analysis, which is one of multiple regression and a researcher-controlled regression method, was used to identify covariates and the magnitude of the relative influence of the independent variables [24]. Following specific assessments were conducted: (1) General characteristics of the subjects were identified, and scores for the nursing work environment, relationship with the head nurse, resilience, and posttraumatic growth are assessed. (2) Differences in the nursing work environment, relationship with the head nurse, resilience, and posttraumatic growth according to the participants’ general characteristics were measured. (3) The relationship between the participants’ nursing work environment, relationship with the head nurse, resilience, and posttraumatic growth was assessed. (4) Factors that affected the participants’ posttraumatic growth were identified.

## 2. Materials and Methods

### 2.1. Setting and Subjects

This study used convenience sampling to select nurses working in the emergency room of a general hospital located in two regions of South Korea (Daegu and Kyungsang-bukdo). The number of subjects required in this study was calculated using G*power 3.1.9.7 program [25] based on linear multiple regression analysis. When the number of predicted samples was set to 3 predictors for the test at the significance level (α) of 0.05, effect size *f*^2^ = 0.15, power (1-β) of 0.80, and 10 total predictors, the appropriate number of subjects was at least 78. Considering the possible dropout rate, 140 emergency department nurses were selected as target subjects. The research subjects were informed of the purpose and procedure of the study and the right to withdraw participation at any point in the study. It was explained that the collected data would not be used exclusively for research purposes and that each participant would receive a small number of gifts even if participation is withdrawn at a later point in the study. A total of 140 questionnaires were distributed, and 13 of them contained incomplete responses. One hundred twenty-seven of them were used for the analysis.

### 2.2. Instruments

#### 2.2.1. Nursing Work Environment

The nursing work environment was measured using the practice environment scale of the nursing work index (PES-NWI). It was developed by Lake [26]; it was adapted to fit Korean participants, and the reliability and validity of the Korean version were verified by Cho et al. [27]. It consists of a total of 29 items across five subscales: nursing participation in hospital affairs (9 items), nursing foundation for quality care (9 items), nursing manager’s leadership, ability, and support for nurses (4 items), adequate staffing and resources (4 items), and collegial nurse–physician relationship (3 items). Each item is rated on a four-point rating scale from “not at all” 1 point to “very much” 4 points, ranging total score from 29 to 116, and an average score of 2.5 points or higher across all items indicates that the nurse agrees that his or her nursing work environment is good; a score less than 2.5 indicates the work environment is not good [11]. Regarding instrument reliability, Cronbach’s α was 0.82 at the time of PES-NWI development [26], 0.93 in the study by Cho et al. [27], and 0.91 in this study.

#### 2.2.2. Relationship with the Head Nurse

The relationship with the head nurse was measured using a tool modified for nurses in Heo and Lee’s study [28] based on the leader–member exchange developed by Scandura and Graen [29]. The tool has 7 items, and each item is rated on a five-point Likert scale from “not so much” 1 point to “very much” 5 points ranging total score from 7 to 35; the higher the score, the better the relationship between the head nurse and the staff nurse. Regarding reliability, Cronbach’s α was 0.86 for the original instrument [29], 0.80 in Heo and Lee’s study [28], and 0.91 in this study.

#### 2.2.3. Resilience

Resilience was measured using the Connor–Davidson resilience scale (CD-RISC) developed by Conor and Davidson [30] and translated by Jung et al. [31] into Korean. The reliability and validity of the Korean Version of the Connor–Davidson resilience scale (K-CD-RISC) were verified by Jung et al. [31]. It has 25 items, and each item is rated on a five-point rating scale ranging from “not true at all” 0 points to “true nearly all of the time” 4 points; total score ranges from 0 to 100, with greater scores indicating higher resilience. The Cronbach’s α was 0.93 for the original tool [30], 0.92 for K-CD-RISC [31], and 0.93 in this study.

#### 2.2.4. Posttraumatic Growth

For posttraumatic growth, the posttraumatic growth inventory (PTGI) was used, developed by Tedeschi and Calhoun [32], and Song et al. [33] translated it into Korean (K-PTGI) and verified its reliability and validity. It has a total of 16 items across four subscales: self-perception (6 items), increase in interpersonal depth (5 items), finding new possibilities (3 items), and increase in spiritual interest (2 items). Each item is rated on a six-point rating scale from “did not experience this change as a result of my crisis” 0 points to “experienced this change to a very great degree as a result of my crisis” 5 points; with ranging total score from 0 to 105; higher the score, more positive were the changes experienced after trauma. The Cronbach’s α was 0.90 at the time of PTGI development [32], 0.94 for K-PTGI [33], and 0.91 in this study.

### 2.3. Data Collection

Data were collected from 13 December 2018, to 15 February 2019. The subjects of the study were 140 nurses working in the emergency room of a general hospital in Daegu and Kyungsangbuk-do who understood the purpose of the study and voluntarily agreed to participate. The questionnaire was self-written and took about 15 min to complete. The completed questionnaire was collected in a sealed envelope.

### 2.4. Data Analysis

The collected data were analyzed using the IBM SPSS Statistics 25 program (IBM, Armonk, NY, USA). The general characteristics of the study subjects, nursing work environment, relationship with the head nurse, resilience, and posttraumatic growth were measured using frequency, percentage, mean, standard deviation, minimum, and/or maximum. The differences in the scores of the nursing work environment, relationship with the head nurse, resilience, and posttraumatic growth according to the general characteristics of the subjects were analyzed using *t*-test and one-way analysis of variance, and then the difference between groups was analyzed using Scheffé’s test. The relationship between the subject’s nursing work environment, relationship with the head nurse, resilience, and posttraumatic growth was analyzed using the Pearson’s correlation coefficient test. Hierarchical multiple regression was used to identify the factors that influenced the subject’s posttraumatic growth.

## 3. Results

### 3.1. General Characteristics of Study Subjects

Demographic characteristics of the subjects are shown in Table 1. Their average age was 29.28 ± 5.75 years. Based on the previous study’s age grouping [34], with 79 (62.2%) aged 20–29 years. Ninety-five (74.8%) had a spouse, and 115 (90.6%) were female. The majority of respondents—92 (72.4%)—had graduated from a four-year college. Forty-two (33.1%) had 2 years or less of a career as a nurse in the emergency department, and 34 (26.8%) had 2 years or more and less than 4 years. As for whether the emergency room was the desired department, 79 (62.2%) answered yes. Sixty-five (51.2%) answered yes to whether they wished to continue working in the emergency room (Table 1).

### 3.2. Scores for Nursing Work Environment, Relationship with the Head Nurse, Resilience, and Post-Traumatic Growth

Scores for the study subjects’ nursing work environment, relationship with the head nurse, resilience, and posttraumatic growth are shown in Table 2. The mean score for nursing work environment was 2.38 ± 0.35 from 1 to 4 points, relationship with the head nurse was 3.05 ± 0.65 from 1 to 5 points, resilience was 3.29 ± 0.48 from 0 to 4 points, and posttraumatic growth was 2.59 from 0 to 5 points.

### 3.3. Differences in Nursing Work Environment, Relationship with the Head Nurse, Resilience, and Post-Traumatic Growth According to General Characteristics

The differences in the nursing work environment, relationship with the head nurse, resilience, and posttraumatic growth according to the general characteristics are shown in Table 3. The nursing work environment showed a statistically significant difference according to age (F = 4.38, *p* = 0.015), career in the emergency department (F = 4.38, *p* = 0.015), whether the emergency department was the desired department (t = 3.60, *p* < 0.001), and whether they wanted to continue working in the emergency department (t = 3.33, *p* = 0.001). After the post hoc test, age did not show statistical significance among age groups. Scheffé’s test showed that subjects with less than 2 years of working experience in the emergency department perceived the nursing work environment more positively than those with 2–4 years. In addition, nurses for whom the emergency room was the desired department and those who wished to continue working in the emergency department reported their nursing work environment as good.

The relationship with the head nurse showed a significant difference according to educational background (F = 4.55, *p* = 0.012), and the duration of working in emergency department (F = 3.77, *p* = 0.013). Those who had an academic background of graduate school or higher reported their relationship with the head nurse as good and those who had 4–6 years of experience working in the emergency department reported a better relationship with the head nurse than those with 2–4 years’ experience. The score for a relationship with the head nurse was also significantly higher for those who wished to continue working in the emergency department (t = 3.58, *p* < 0.001).

Resilience scores were significantly higher for those whom the emergency department was the desired unit to work (t = 2.46, *p* = 0.015) and those who wished to continue working in the emergency room (t = 2.35, *p* = 0.021).

Posttraumatic growth scores showed none of the statistically significant differences according to the general characteristics.

### 3.4. Correlations between Variables

With respect to correlations between variables, the relationships between nursing work environment and relationship with the head nurse (r = 0.47, *p* < 0.001), between nursing work environment and resilience (r = 0.26, *p* = 0.003), between nursing environment and post-traumatic growth (r = 0.32, *p* < 0.001), between resilience and relationship with the head nurse (r = 0.24, *p* = 0.008), between post-traumatic growth and relationship with the head nurse (r = 0.37, *p* < 0.001), and between resilience and post-traumatic growth (r = 0.65, *p* < 0.001) showed statistically significant positive relationships (Table 4).

### 3.5. Hierarchical Regression Analysis of Predictors Associated with Post-Traumatic Growth

The results of a hierarchical multiple regression analysis of the relationship among nursing work environment, relationship with the head nurse, and resilience to identify the variables that affected the subjects’ posttraumatic growth are shown in Table 5.

First, to test the regression analysis assumption for the independent variable, we checked for multicollinearity. The tolerance limit (tolerance) was 0.66–0.92, which was more than 0.1, and the variance inflation factor (VIF) was 1.00–1.53, which were all less than 10. The correlation between the independent variables was also 0.24–0.65, which was less than 0.80, confirming that there was no problem of multicollinearity between the independent variables.

In the first stage, nursing work environment (β = 0.32, *p* < 0.001) was entered and found that it was a significant factor in post-traumatic growth (F = 3.72, *p* < 0.001), accounting for 9.3% of the variance (model 1). In model 2, wherein relationship with the head nurse was added to model 1 controlling for the nursing work environment, relationship with the head nurse (β = 0.29, *p* = 0.003) was found to have a significant effect on posttraumatic growth, explaining 5.8% of posttraumatic growth. In model 3, resilience (β = 0.59, *p* < 0.001) was found to have a significant effect on posttraumatic growth and accounted for 31.9% of the variance, controlling for the nursing work environment and relationship with the head nurse. All significant variables were found to account for 47.0% of posttraumatic growth.

## 4. Discussion

This study attempted to provide basic data to establish intervention programs for posttraumatic growth in emergency department nurses by understanding the effects of emergency room nurses’ work environment, relationship with the head nurse, and resilience on posttraumatic growth.

In this study, the average score for the nursing work environment perceived by emergency department nurses was 2.38 points. In previous studies, the average score for the nursing work environment was 2.85 in the integrated nursing unit and 2.45 in the medical or surgical unit [35], and 2.51 in three university hospitals [36]. In a previous study conducted with emergency department nurses [37], the average score of the nursing work environment was 2.32, indicating that the score in our study was higher. However, the above scores indicate that emergency department nurses reported lower scores for their nursing work environment, which was lower than nurses in other units. Eileen et al. [11] reported that nurses thought their nursing work environment was good at the theoretical midpoint value of 2.5 points or higher. Considering this, it seems necessary to make efforts to improve the nursing work environment of the emergency department.

The average score for a relationship with the head nurse was 3.05, which was lower than other previous studies, compared to 3.51 in a small and medium-sized general hospital [38], and 3.09 in Heo and Lee’s [28] study. Environment for active communication with head nurses should be fostered, and an educational program for head nurses and staff nurses to increase positive relationships should be developed.

The average score for resilience in this study was 3.29 points, compared to 3.59 points in Park and Lee’s [39] study on the resilience of emergency room nurses using the same tool, and 3.60 points in Moon’s [40] study. In Park and Lee’s study [39], a clinical experience of more than 5 months among emergency room nurses may have affected the score. It is necessary to provide program interventions that can improve resilience so that nurses can respond to crisis situations and perform various tasks efficiently.

The average posttraumatic growth score was 57.40, and the item average was 3.59 in this study. This is lower than the score of 78.1 among healthcare workers at a large academic center in the Southeastern United States [7]. Although this may involve cultural differences, the subjects of our study were nurses working in the emergency department, and it can be assumed that they were more likely to experience posttraumatic stress. This can be confirmed by a future study on the relationship between posttraumatic stress and posttraumatic growth in emergency department nurses.

A hierarchical regression analysis was performed to determine whether variables such as nursing work environment, relationship with the head nurse, and resilience are associated with posttraumatic growth. In model 1, nursing work environment (β = 0.32, *p* < 0.001) explained 9.3% of post-traumatic growth. No prior study has found the impact of nursing work environment on posttraumatic growth; the nursing work environment affects the degree of posttraumatic growth experienced by nurses, and hence, assessing nurses’ perception of their work environment will be an important intervention.

In model 2, after introducing the variable of relationship with the head nurse while controlling for the nursing work environment, it was found that relationship with the head nurse (β = 0.29, *p* = 0.003) explained 5.8% of the posttraumatic growth variable. Although there are no previous studies examining the effect of the relationship with the head nurse on posttraumatic growth, a similar context of social support has been investigated in previous studies. In a study on the relationship between social support and posttraumatic growth in psychological nurses, social support was found to have a positive effect on posttraumatic growth [34]. Considering that emergency department nurses often encounter urgent situations in their workplace and considering the results of previous studies that show social support as a factor with significant influence on posttraumatic stress disorders [2,8], it can be said that relationship with the head nurse is a very important factor.

In model 3, when controlling for other variables, resilience (β = 0.59, *p* < 0.001) was a positive influencing factor for posttraumatic growth, explaining 31.9% of the variance and all variables were found to explain posttraumatic growth by 47.0%. This indicates that resilience is the most influential predictor of posttraumatic growth, and this can be said to be related to previous findings that high resilience at work is related to high work performance of nurses [20]. Planning and implementing programs that can improve resilience are necessary for emergency department nurses.

This study has some limitations. First, the subjects were local emergency department nurses limited to a city in South Korea. Therefore, the results have limited generalizability; it is necessary to expand the number of samples to more areas, male nurses and nurses working in various nursing departments. Second, since this study used cross-sectional data, the results cannot be used to infer causal relationships. Therefore, further research using longitudinal data should be facilitated. Third, based on a previous study [18], resilience has multisystemic nature, and further study should be performed, including the systemic and culturally sensitive risk exposure, promotive and protective processes, and outcomes [41]. Finally, in Ungar [23]’s study, posttraumatic growth was one of the resilience patterns and the optimal level of functioning of resilience and very sensitive to systemic factors. Therefore, the research should be performed the resilience as an outcome variable and with cultural sensitivity.

The results of this study show that it is necessary to develop a program that can increase posttraumatic growth through improvement in the work environment, increase in resilience, and training to improve the relationship with the head nurse, and research to verify the effectiveness of the program needs to be conducted. Since a gap in knowledge, skills and competency among healthcare professionals in the management of disasters exists, standardization of a program with a pre-training evaluation should be developed for nurses working in different units [42]. Further study is necessary whether studied variables show the potential mediation or interaction effects. This study is meaningful in that it tries to identify the related factors of posttraumatic growth in emergency room nurses. Further studies are suggested that identifying the main factors affecting posttraumatic growth in Korean emergency room nurses.

## 5. Conclusions

This study tried to provide the data for identifying the association of nursing work environment, relationship with the head nurse, and resilience with posttraumatic growth in emergency department nurses. The results showed that the most influential factor on the posttraumatic growth of emergency room nurses was resilience. Although there are several limitations in this study, the findings showed it is necessary to prepare a program that can improve resilience and relationship with the head nurse especially considering Korea’s hierarchical culture with seniors. Based on the results of this study, it is necessary to identify the nursing work environment and posttraumatic growth in emergency department nurses in various areas. It is also necessary to expand the number of samples to more areas, male nurses, and nurses working in various nursing departments to be able to generalize the results of the study and to identify the causal relationship the longitudinal analysis should be performed.

## Figures and Tables

**Table 1 ijerph-18-02857-t001:** General characteristics of the subject (*n* = 127).

Variables	Categories	*n* (%) or M ± SD
Age	(M ± SD)	29.28 ± 5.75
	20–29	79 (62.2)
	30–39	42 (33.1)
	≥40	6 (4.7)
Spouse	Yes	32 (25.2)
	No	95 (74.8)
Gender	Female	115 (90.6)
	Male	12 (9.5)
Education	College	26 (20.5)
	University	92 (72.4)
	Graduate and more	9 (7.1)
Career in ED	<2	42 (33.1)
	2–<4	34 (26.8)
	4–<6	27 (21.3)
	≥6	24 (18.9)
ED is a Wanted unit	Yes	79 (62.2)
	No	48 (37.8)
Continuously wanted working in ED	Yes	65 (51.2)
	No	62 (48.8)

ED = emergency department.

**Table 2 ijerph-18-02857-t002:** Degrees of the nursing work environment, relationship with the head nurse, resilience, and posttraumatic growth. (*n* = 127).

Variables	M ± SD	Possible Range	M/Item ± SD	Possible Range (Item)
NWE	69.02 ± 10.17	41–96	2.38 ± 0.35	1.41–3.31
RHN	21.33 ± 4.54	7–33	3.05 ± 0.65	1.00–4.71
Resilience	82.26 ± 11.94	48–108	3.29 ± 0.48	1.92–4.32
PTG	47.40 ± 10.26	20–72	2.59 ± 0.64	0.88–4.13

NWE = nursing work environment; RHN = relationship with head nurse; PTG = post-traumatic growth.

**Table 3 ijerph-18-02857-t003:** Differences in the work environment, relationship with the head nurse, resilience, and posttraumatic growth according to the general characteristics.

Variables	Categories	Nursing Work Environment		Relationship with Head Nurse		Resilience		Posttraumatic Growth	
		M ± SD	t/F (*p*) Scheffé	M ± SD	t/F (*p*) Scheffé	M ± SD	t/F (*p*) Scheffé	M ± SD	t/F (*p*) Scheffé
Age	20–29	2.43 ± 0.34	4.38 (0.015)	3.01 ± 0.57	2.74 (0.069)	3.28 ± 0.53	1.84 (0.163)	3.57 ± 0.63	2.49 (0.087)
	30–39	2.26 ± 0.33		3.02 ± 0.75		3.25 ± 0.38		3.54 ± 0.63	
	≥40	2.57 ± 0.38		3.64 ± 0.71		3.65 ± 0.26		4.15 ± 0.71	
Spouse	Yes	2.32 ± 0.29	−1.06 (0.291)	3.13 ± 0.73	0.78 (0.435)	3.33 ± 0.41	0.54 (0.590)	3.75 ± 0.57	1.69 (0.094)
	No	2.40 ± 0.37		3.02 ± 0.62		3.28 ± 0.50		3.53 ± 0.66	
Gender	Female	2.38 ± 0.35	−0.47 (0.641)	3.01 ± 0.66	−1.76 (0.082)	3.27 ± 0.47	−1.71 (0.089)	3.58 ± 0.63	−0.62 (0.533)
	Male	2.43 ± 0.33		3.36 ± 0.44		3.51 ± 0.48		3.70 ± 0.80	
Education	College ^a^	2.37 ± 0.37	0.04 (0.964)	2.87 ± 0.62	4.55 (0.012)	3.18 ± 0.50	1.85 (0.161)	3.51 ± 0.56	0.76 (0.472)
	University ^b^	2.38 ± 0.34		3.04 ± 0.62	a,b < c	3.30 ± 0.48		3.59 ± 0.66	
	Graduate and more ^c^	2.41 ± 0.45		3.60 ± 0.72		3.53 ± 0.39		3.82 ± 0.73	
Career in ED	<2 ^a^	2.49 ± 0.34	3.86 (0.011)	3.04 ± 0.55	3.77 (0.013)	3.35 ± 0.52	1.42 (0.239)	3.63 ± 0.71	0.92 (0.435)
	2–<4 ^b^	2.40 ± 0.32	a > b	2.77 ± 0.55	c > b	3.16 ± 0.51		3.45 ± 0.55	
	4–<6 ^c^	2.30 ± 0.32		3.28 ± 0.51		3.28 ± 0.47		3.71 ± 0.60	
	≥6 ^d^	2.27 ± 0.37		3.19 ± 0.91		3.38 ± 0.34		3.58 ± 0.67	
ED is a Wanted unit	Yes	2.46 ± 0.36	3.60 (<0.001)	3.11 ± 0.67	1.50 (0.137)	3.37 ± 0.46	2.46 (0.015)	3.63 ± 0.66	1.02 (0.309)
	No	2.24 ± 0.29		2.94 ± 0.60		3.16 ± 0.48		3.51 ± 0.60	
Continuously wanted working in ED	Yes	2.48 ± 0.36	3.33 (0.001)	3.24 ± 0.60	3.58 (<0.001)	3.39 ± 0.48	2.35 (0.021)	3.66 ± 0.71	1.27 (0.206)
	No	2.28 ± 0.31		2.85 ± 0.64		3.19 ± 0.45		3.51 ± 0.55	

ED = emergency department; ^a^, ^b^, ^c^, ^d^: the results of Post-hoc test, Scheffe.

**Table 4 ijerph-18-02857-t004:** Relationship between variables.

Variables	Nursing Work Environment	Relationship with Head Nurse	Resilience
	r (*p*)		
Relationship with the head nurse	0.47 (<0.001)		
Resilience	0.26 (0.003)	0.24 (0.008)	
Post-traumatic growth	0.32 (<0.001)	0.37 (<0.001)	0.65 (<0.001)

**Table 5 ijerph-18-02857-t005:** Hierarchical regression analysis of predictors associated with posttraumatic growth (*n* = 127).

Model	Variables	B	S.E	β	t	*p*	*Tolerance*	*VI* *F*	*Adj. R* ^2^	*F (p)*
Model 1	(constant)	35.41	5.97		5.93	<0.001			0.09	13.84
	NWE	0.32	0.09	0.32	3.72	<0.001	0.92	1.00		(<0.001)
Model 2	(constant)	30.95	5.96		5.19	<0.001			0.15	12.16
	NWE	0.18	0.09	0.18	1.94	0.054	0.78	1.28		(<0.001)
	RHN	0.65	0.21	0.29	3.09	0.003	0.78	1.28		
Model 3	(constant)	1.24	5.82		0.21	0.832			0.47	38.20
	NWE	0.07	0.08	0.07	0.93	0.354	0.66	1.53		(<0.001)
	RHN	0.45	0.17	0.20	2.71	0.008	0.71	1.40		
	Resilience	0.51	0.06	0.59	8.70	<0.001	0.89	1.12		

VIF = variance inflation factor; NWE = nursing work environment; RHN = relationship with head nurse.

## Data Availability

Data available on request due to ethical considerations; the data presented in this research are available on request from the corresponding author.

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
