# Peer review of "Association of Nursing Work Environment, Relationship with the Head Nurse, and Resilience with Post-Traumatic Growth in Emergency Department Nurses"

_ijerph, 2021, doi:10.3390/ijerph18062857_

Round 1
Reviewer 1 Report
With the frequent emergence of new infectious diseases, it is considered to be a very significant study these days with a high possibility of trauma to emergency room-related health care professionals. If the following contents are revised, it is expected to be more meaningful research.
First of all, the reason for choosing hierarchical regression analysis should be presented in the introduction.
Second, in the case of the methods, the power (.95) is too high when calculating the sample size. Please lower it below .85 and present the sample size recalculated accordingly.
In line 121, the author's name is given by first name, not last name. I hope this typo could be corrected.
Among the research results, please provide the basis for age classification by citing the previous studies.
You need to edit the spacing between lines 231 and 232.
In lines 328-338, it is necessary to add, compare, and describe previous studies that can support the effects of the intervention on post-traumatic growth, the main concept of this study in the discussion.
In terms of the format, if some contents (for example, the arrangement of statistics in the table) are modified according to the submission guidelines, it is considered that the research will be more significant.
Thank you for your efforts.
Author Response
Comments: First of all, the reason for choosing hierarchical regression analysis should be presented in the introduction.
Response: Thank you for your comments. I described in line 92-95.
Comments: Second, in the case of the methods, the power (.95) is too high when calculating the sample size. Please lower it below .85 and present the sample size recalculated accordingly.
Response: Thank you for your comments. I set the power of .80, recalculated the sample size, and described in line 110-111.
Comments: In line 121, the author's name is given by first name, not last name. I hope this typo could be corrected.
Response: Thank you for your comments. I corrected it and it is now in line 124.
Comments: Among the research results, please provide the basis for age classification by citing the previous studies.
Response: Thank you for your comments. I cited the age group in line 183. Based on the previous study, I regroup the age. However, the result showed that the age is not a statistically significant variable for regression analysis anymore, so I analyzed and revised the whole result section.
Comments: You need to edit the spacing between lines 231 and 232.
Response: Thank you for your comments. I edited the spacing and now it is in line 260.
Comments: In lines 328-338, it is necessary to add, compare, and describe previous studies that can support the effects of the intervention on post-traumatic growth, the main concept of this study in the discussion.
Response: Thank you for your comments. I deleted the whole sentences from previous lines 328-334 according to the reviewer 2’s suggestion.
Comments: In terms of the format, if some contents (for example, the arrangement of statistics in the table) are modified according to the submission guidelines, it is considered that the research will be more significant.
Response: Thank you for your comments. I modified the contest according to the submission guidelines.
Reviewer 2 Report
Thank you very much for the opportunity to read this manuscript. It studies predictors of posttraumatic growth in emergency department nurses. Overall, the manuscript has been well prepared. My major critique relates to the conceptualization of resilience. The authors have assessed a nice set of variables that fit very well with the most recent conceptualization of resilience. There is no clear underlying theory and hypotheses are missing. I also wonder about the study design in the light of the theoretical background since factors have not been assessed that would be critical to properly study posttraumatic growth. The results section is often redundant.
Introduction
I would like to invite the authors to restructure and expand their introduction with the following state-of-the-art facts of resilience science:
- Resilience is the ability for positive development and functionality under atypical stress which needs protective and promotive processes and factors.
- Resilience is currently conceptualized as a multisystemic construct: this means that the old tradition of seeing resilience merely as an internal trait of blessed individuals (what the CD RISC measures) is outdated. “Resilience” refers to processes between different promotive and protective factors of multiple systems that go far beyond psychological variables (such as physiology, physical environment, social-ecological environment, etc.) and the authors have assessed a nice variety that would fit this conceptualization: work environment which relates to social relationships, roles, responsibilities, leadership; relationship to head nurse; CD-RISC for trait like resilience; education; motivation or person-environment fit,…
- How can posttraumatic growth be related to resilience: The very underlying process of resilience is adaptation. Stressed individuals need resources (traits, social support, etc.) for adaptation. What does adaptation mean? A change. The thinking of resilience as “bouncing back” has received severe critique and this was one of the foundations of the CD-RISC. There is no bouncing back after a trauma, a trauma is defined as cutting life into a before and after. Posttraumatic growth has been conceptualized by Tedeschi and Calhoun themselves as a means to become more resilient (See: The posttraumatic growth workbook). So, posttraumatic growth should be seen as “bouncing forward” in a sense. Posttraumatic growth is a process of resilience and in order to occur certain resources are necessary which depend on the context.
- So, the CD-RISC only assesses a certain set of resilience-supporting resources, but not resilience itself and it is crucial to respect this in the manuscript.
- Work by Michael Ungar brings many of those facts to the point (see also Ann Masten):
- https://doi.org/10.5751/ES-10385-230434
- https://doi.org/10.1016/S2215-0366(19)30434-1
- https://doi.org/10.1111/jmft.12124
- https://doi.org/10.1016/j.chiabu.2019.104098
- https://doi.org/10.1111/cdev.12205
- Overall, if the authors use the theory of multisystemic resilience as a foundation for their manuscript and define posttraumatic growth as a process of resilience, it would much improve and give the paper a stronger basis.
Materials and Methods
- Based on the background and general literature on posttraumatic growth I wonder why posttraumatic stress and an indicator for stress load, traumatic experiences, etc. have not been used in the study? Are there any such variables in the dataset?
- 2.1 study design: that is not a study design that is presented here. The study design should be about the data collection/procedure and parts of the section on setting and subjects.
- 2.2 it says that 140 questionnaires were distributed, but only 127 were used. What happened to the other 13?
- 2.3. please provide a sample item and the range for each instrument
- 2.3. describe the covariates and give reasons for (1) why these age groups were used, (2) how did you decide on the groups for “Career in registered nurse” and “Career in ED”.
- 2.5: Why Scheffe test and not one of the others? What is its advantage for this context?
Results:
- Table design is not coherent. I would suggest using APA guidelines.
- Tables are not needed if everything is written in the text. You either use a table and give a very brief description in the text, or you put all the numbers into the text and do not use a table. Here, mostly everything is in the text that is also in the tables which is redundant.
- Table 3: there are sometimes a, b, c,.. in the categories column but there is no description in the notes.
- 3.5: even though the authors did a pre-test to see which covariates show significant differences in posttraumatic growth and thus the authors only included age in the regression model, I would still ask the authors to include all their variables into the model which would be more appropriate from a multisystemic resilience perspective.
- Optionally: maybe the authors might want to think about potential moderation/interaction effects to give their analysis some more complexity.
Discussion:
- Please frame the discussion in a way that all these results relate to South Korea. It is written way too generally. Resilience research places a strong focus on contextual and cultural specificities.
- “Changes in nursing organizational culture are needed through education on positive interpersonal communication.”: this cannot be stated based on the results, or what does the measure for relationship with head nurse assess?
- “that older age was a statistically significant factor in post-traumatic growth. Thus, our result supports the evidence that post-traumatic growth is acquired through the experience of trauma, and it can be inferred that older the age, more are the opportunities to grow.”: The study does not say that posttraumatic growth is acquired through the experience of trauma since this has not been assessed and tested!
- Lines 328-334: it has not been assessed if “resilience” can change situational demands or a stressful environment. -> delete. Also, the statement about a mindfulness program: this study was done in the USA -> culture! Second, context-specificity of resources: only because this intervention helped for burn-out must not mean that it helps for posttraumatic growth. So, this whole section cannot be used in the context of this study.
- The part on Corona: misplaced. If the authors really want to mention something about Corona in their article, which is not necessary, it should rather belong into the introduction as a means for the relevance of this study.
- Limitations: There is essentially only one limitation named by the authors. the authors should at least additionally discuss the limitations of using cross-sectional data, and the missing of crucial variables that should be assessed in any study on posttraumatic growth.
- Limitations: also, how do the authors know that the variables that they’ve assessed are “the main factors affecting posttraumatic growth in emergency room nurses”? That is simply not true. In order to make any claim like that, they should have done a qualitative study with their study population first in order to identify such factors for this specific context in this specific culture.
- Limitations: I don't understand the necessity to develop "department-specific tools". Even though I get the message and see the value in such an endeavor, the data and results do not point to this implication. I'd delete this since there is no foundation for this statement. This would have needed a qualitative sub-study on posttraumatic growth to compare the qualitatively identified PTG factors with the ones that are assessed with the quantitative PTG measure.
- conclusion: The stated aim of the study ("developing a posttraumatic growth program") is not present in the introduction.
- conclusion: there is redundancy.
Author Response
Thank you for your comments for improving our manuscript. Please see the attachment

Reviewer 3 Report
Firstly, I would like to congrats the authors for the opportunity of reading and reviewing their manuscript.
The research is well conducted.
The paper is a descriptive study and basically presents the results of a survey-based research among nurses.
The research is well designed and well-conducted, and the results are properly presented.
I have some suggestions:
In the discussion, you should also discuss the gaps that arise from teaching and training problems.
It is important, because the conclusions you presented are accurate, but they must have a stronger background.
As an active physician, I see a nurse's problem every day and it is very important that they receive more drills and training, but we will not do this without changing the training and education programs.
Please read following articles to see more about this problem:
https://doi.org/10.1186/s12889-021-10165-5
Regarding the references, it is convenient that they are reviewed. Some are incomplete, and many of them do not have the doi. You should also expand this section adding more international references.
Good luck, I really enjoyed reading this paper.
Author Response
I deeply appreciate your comments for improving our manuscript.
Comments: In the discussion, you should also discuss the gaps that arise from teaching and training problems.
Response: Thank you for your comment. I described the gaps in line 341-344.
Comments: It is important, because the conclusions you presented are accurate, but they must have a stronger background.
Response: Thank you for your comment. I described in the conclusion section.
Comments: As an active physician, I see a nurse's problem every day and it is very important that they receive more drills and training, but we will not do this without changing the training and education programs.
Please read following articles to see more about this problem:
https://doi.org/10.1186/s12889-021-10165-5
Response: Thank you for your comment. I used the research article that you suggested and described in the discussion section.
Comments: Regarding the references, it is convenient that they are reviewed. Some are incomplete, and many of them do not have the doi. You should also expand this section adding more international references.
Response: Thank you for your comment. I inserted doi.